**Data Availability Statement:** Interested researchers may submit a data use proposal to MAPS (askmaps@maps.org) to gain access to the underlying de-identified data. Following internal

# The cost-effectiveness of MDMA-assisted psychotherapy for the treatment of chronic, treatment-resistant PTSD

**Elliot Marseille**[1,2]*, **James G. Kahn**[2], **Berra Yazar-Klosinski**[3], **Rick Doblin**[3]

**1** Health Strategies International, Oakland, California, United States of America, **2** University of California, San Francisco, California, United States of America, **3** Multidisciplinary Association for Psychedelic Studies (MAPS), Santa Cruz, California, United States of America

* emarseille@comcast.net

## Abstract

### Background

Chronic posttraumatic stress disorder (PTSD) is a disabling condition that generates considerable morbidity, mortality, and both medical and indirect social costs. Treatment options are limited. A novel therapy using 3,4-methylenedioxymethamphetamine (MDMA) has shown efficacy in six phase 2 trials. Its cost-effectiveness is unknown.

### Methods and findings

To assess the cost-effectiveness of MDMA-assisted psychotherapy (MAP) from the health care payer's perspective, we constructed a decision-analytic Markov model to portray the costs and health benefits of treating patients with chronic, severe, or extreme, treatment-resistant PTSD with MAP. In six double-blind phase 2 trials, MAP consisted of a mean of 2.5 90-minute trauma-focused psychotherapy sessions before two 8-hour sessions with MDMA (mean dose of 125 mg), followed by a mean of 3.5 integration sessions for each active session. The control group received an inactive placebo or 25–40 mg. of MDMA, and otherwise followed the same regimen. Our model calculates net medical costs, mortality, quality-adjusted life-years (QALYs), and incremental cost-effectiveness ratios. Efficacy was based on the pooled results of six randomized controlled phase 2 trials with 105 subjects; and a four-year follow-up of 19 subjects. Other inputs were based on published literature and on assumptions when data were unavailable. We modeled results over a 30-year analytic horizon and conducted extensive sensitivity analyses. Our model calculates expected medical costs, mortality, quality-adjusted life-years (QALYs), and incremental cost-effectiveness ratio. Future costs and QALYs were discounted at 3% per year. For 1,000 individuals, MAP generates discounted net savings of $103.2 million over 30 years while accruing 5,553 discounted QALYs, compared to continued standard of care. MAP breaks even on cost at 3.1 years while delivering 918 QALYs. Making the conservative assumption that benefits cease after one year, MAP would accrue net costs of $7.6 million while generating 288 QALYS, or $26,427 per QALY gained.

review, MAPS may contact the requesting
researcher with clarifying questions. This results in
a Data Use Agreement and release of the data.

**Funding:** The author(s) received no specific
funding for this work.

**Competing interests:** The authors have declared
that no competing interests exist.

## Conclusion

MAP provided to patients with severe or extreme, chronic PTSD appears to be cost-saving
while delivering substantial clinical benefit. Third-party payers are likely to save money
within three years by covering this form of therapy.

## Background

Posttraumatic Stress Disorder (PTSD) is a serious psychiatric condition that may follow a trau-
matic event. It is characterized by symptoms including anxiety, depersonalization, derealiza-
tion, insomnia, and recurring nightmares. These are often sufficiently debilitating that normal
work and social activities are impaired or impossible [1]. PTSD is accompanied by an elevated
risk of mood-related co-morbidity and mortality including depression, suicidal ideation, and
completed suicide [2–4]. It can also cause stress-mediated physical health problems such as
cardiovascular disease and type-2 diabetes [5], alcohol abuse, high caloric intake and BMI, and
smoking [6–8]. Early indications suggest that COVID-19 may cause PTSD in young people
[9]. In addition to burden of disease, PTSD generates substantial medical care costs in the U.S.
including $44.3 billion (2019 dollars) for hospitalization between 2002–2011 [1].

Interviews of a nationally-representative sample of 9,282 American adults conducted
between February 2001 and April 2003 indicated a lifetime prevalence of PTSD of 6.8% [10] and
past-year prevalence of 3.5% [11]. These results are similar to an earlier survey that found a life-
time prevalence of 7.8%. In both surveys women were more than twice as likely as men to suffer
from PTSD [12]. Using the past-year prevalence, roughly 11.8 million American adults are cur-
rently affected by PTSD. While many individuals with PTSD experience remission with or with-
out treatment, a large portion, perhaps 50%, experience recurring or chronic PTSD according to a
2015 systematic review [13]. This is consistent with the estimated 40–60% of patients who in sepa-
rate analyses responded inadequately to standard pharmacotherapies [14–17].

Against this backdrop, recent evidence of benefit from a novel treatment, MDMA-assisted
psychotherapy (MAP), is particularly relevant. The mechanisms of MDMA's action and its
particular efficacy for resolving PTSD are fairly well understood and described elsewhere [13,
18, 19].

The non-profit research and educational organization, the Multidisciplinary Association of
Psychedelic Studies (MAPS), is working with the Food and Drug Administration (FDA) and
the European Medicines Agency to build on pooled efficacy data from six phase 2 trials. MAPS
has received FDA "Breakthrough Therapy" designation for MAP, with two phase 3 trials
underway [20]. This suggests the possibility that MDMA, currently designated a DEA Sched-
ule 1 drug of abuse, when combined with psychotherapy, may become a legal, licensed treat-
ment available by prescription within 24–36 months [18].

The magnitude of the unmet public health need coupled with promising results from phase
2 trials argues for advanced preparation for rapid access following FDA approval. Third-party
payers are unlikely to adopt MAP as a benefit absent information on costs and cost-effective-
ness. This analysis addresses this need.

## Methods

### Overview

We developed a decision analytic model to portray clinical benefits, MAP costs, and net medi-
cal costs in a cohort of 1,000 patients reflecting the age and PTSD severity of 105 patients

treated in six phase 2 clinical trials conducted between 2004 and 2017 [17]. Based on the Clinician-Administered PTSD Scale per DSM-IV (CAPS-IV) patients were classified and portrayed in a Markov process using standard criteria for: asymptomatic, mild, moderate, severe, and extreme PTSD [21]. Individuals in these trials had failed at least one prior treatment with standard interventions. MAP efficacy is portrayed as a change in the distribution by severity category at the trials' endpoints compared with baseline. (S1 Fig in S1 File). Mortality, health state utilities, and medical costs were estimated from published literature and vary by severity category. The Markov simulation is annual until death, with costs and QALYs discounted to the present at 3% annually, and with results presented for several time horizons (Fig 1).

## Patient population

The 105 subjects of the six double-blind controlled pilot studies (31 control; 74 active group) suffered moderate to extreme chronic PTSD with an average duration of 197.9 months (SD, 139.1) in the controls and 222.6 months (SD, 208.5) in the treatment group. The mean age was 40.5 years (SD, 10.6). Females constituted 58.1% and whites 87.6%. Mean CAPS scores at baseline were 81.3 (SD, 15.9) in controls (9.7%, 38.7% and 51.6% with moderate, severe, and extreme PTSD, respectively) and 85.8 (SD, 19.3) in the treatment group (12.2%, 31.1% and 56.8%). All subjects had failed at least one conventional therapy for PTSD. Further details on patients are published elsewhere [17].

## Treatment protocol

Following recruitment, randomization, and two to three non-drug 90-minute therapy sessions, participants received blinded doses of placebo/control (0 mg placebo; 25 mg, 30 mg, or 40 mg MDMA) or active (larger) doses of MDMA (75, 100, or 125 mg) administered during two 8-hour psychotherapy sessions conducted 3–5 weeks apart. The initial dose was followed 1.5–2.5 hours later by an optional supplemental half dose. On the day following each active session, participants received a 90-minute integration session. Two or three additional integration sessions followed within one month. Details on recruitment, screening and treatment are provided elsewhere [17].

## Representation of clinical trial results

All simulations in the Markov model incorporate the aggregated empirical results of the six MAPS trials, at 1–2 months following the second experimental session. Patients transitioned to death according to the mortality associated with each severity category. In the base-case analysis, patients in both control and treatment conditions remain in the same severity category achieved at the end of the trial. Thus, the intervention effect is captured as the difference in PTSD severity following the intervention versus baseline. This is consistent with a 3.54-year follow-up study of 19 subjects who completed the phase 2 trials: a modest though statistically insignificant *improvement* in PTSD was reported at a mean of 45.4 months (SD,17.3) following the initial improvement associated with MAP [22]. No improvement or remission in the control group is also consistent with the stable clinical status of patients who, by trial inclusion criteria, suffer from chronic PTSD [22]. Further, limited data available on the long-term trajectory of PTSD suggest that spontaneous remission is predominantly confined to the first few years following diagnosis, and that after seven years, the 50% of patients without remission experience chronic (stable or recurrent) PTSD [13, 23, 24].

The control condition in the phase 2 trials does not represent a feasible, real-world treatment option since it consists of psychotherapy combined with either a placebo (two trials) or a 25–40 mg dose of MDMA (four trials) believed to be clinically inactive [18]. In the base-case

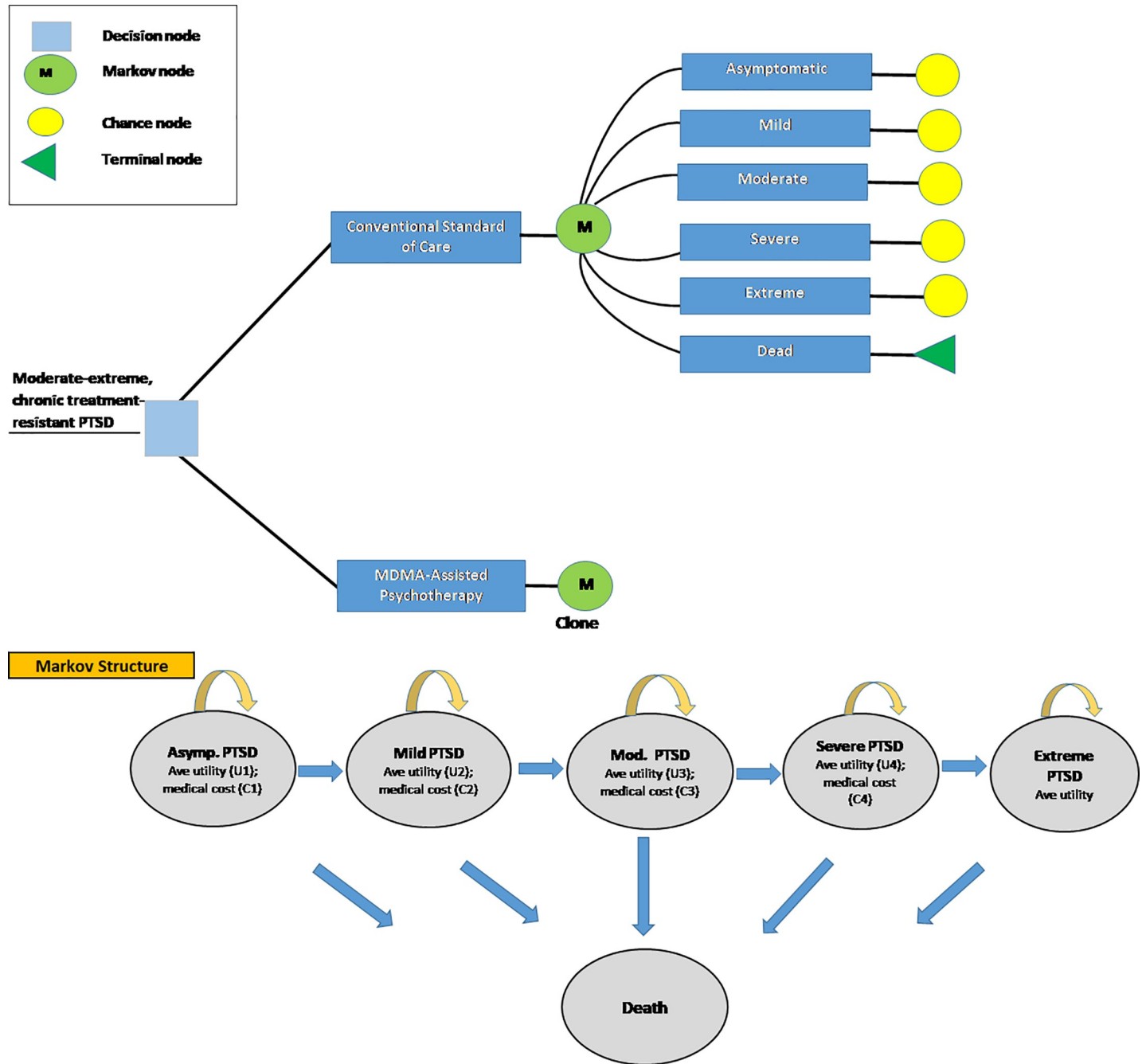

**Fig 1. Decision tree and Markov structure.** Patients are randomly assigned to continuing conventional standard of care or to MDMA-assisted psychotherapy (MAP). At the time of trial completion, a Markov process begins. After each annual cycle patients either remained in the same severity category, progress to a more severe form of PTSD or transition to death (terminal node). Transition probabilities designated by the arrows in the Markov structure, health state utilities, and annual medical care costs are specific to each of the severity categories.

analysis, we therefore modeled the costs and benefits of the active treatment group after receiving MAP with the same group at baseline, i.e., as if they had not received MAP. Because those in the control condition experienced some improvement, in a sensitivity analysis an additional comparison was implemented, namely of patients in the active treatment arm versus controls. (See, "Alternative comparison" and "Further evidence of MAP effectiveness" in S1 File).

The model was implemented in Excel® (Office 365, Microsoft Corporation) and used @RISK® (Palisade Corporation, version 7.6.1) software for sensitivity analyses. Future costs and QALYs were discounted at 3% per year and all costs were inflated to 2019 using the Bureau of Labor Statistics medical care Consumer Price Index [25].

## Health state utility values

Utilities were assigned to mild, moderate, and severe PTSD according to estimates for mild, moderate, and severe anxiety from the Global Burden of Disease, of 0.97, 0.851 and 0.477, respectively [26]. For extreme PTSD we assigned a value of 0.369 based on the administration of the Quality of Well Being-Self-Administered Scale in a cohort of U.S. veterans with a mean CAPS score of 122.1 [27]. Adverse events associated with the trial were transient and managed with benzodiazepines or sleep aids, and are not reflected in the current analysis [17].

## Intervention costs

We used micro-costing, in which each resource needed to deliver MAP was identified, assigned a unit cost via CPT code and private or public payer price schedule, and summed. Research activities were excluded. (See, "Estimation of intervention costs" in S1 File). See Table 1.

## Medical care costs

We estimated the medical costs of patients with PTSD from four studies with five separate estimates [28–31]. To combine the studies, we weighted them by the square root of sample sizes (reflecting statistical variance) to yield an average annual medical cost of $19,899, which we assume to be the cost associated with severe PTSD. (See, "Estimation of PTSD-related medical care costs" and S2 and S3 Tables in S1 File). Extreme PTSD was assumed to be 20% higher than severe PTSD, moderate to be 75% lower, and mild to be 50% of severe (subjected to sensitivity analyses). Asymptomatic cases cost $4,949 annually [28]. See Table 1.

PTSD is associated with higher mental health and general medical care costs, and costs increase with severity [32]. As patients experience reductions in PTSD symptoms, it is unrealistic to assume an immediate reduction in health care expenditure, particularly for chronic conditions such as type 2 diabetes or cardiovascular disease. Thus, the model conservatively distributes the reduction in health care costs over five years: no cost reduction in the first year following MAP, and 25% of the reduction in each successive year.

## Mortality

PTSD is associated with elevated mortality, implemented in the model as relative risk by severity category using age-specific background U.S., mortality rate as the referent. In a study of 637 veterans (12.2% female) all-cause relative mortality risk for PTSD patients was 2.28 [8]. Because we are aware of no mortality risk data stratified by severity, we set the relative risk at 10% above this level for severe and an additional 10% higher for extreme PTSD, moderate PTSD at 10% below severe, and mild an additional 15% below that. The resulting relative mortality risks are 2.76, 2.51, 2.05, and 1.74, respectively.

## Analytic time horizon

Our analysis projects MAP benefits beyond the 4 years to which empirical data are confined. The base-case analysis assesses consequences out to 30 years, assuming retention of clinical

**Table 1. Model input values, ranges for sensitivity analyses, distributions, and data sources.**

| | Key inputs | Value | Standard deviation or range | Distribution | Source |
|---|---|---|---|---|---|
| **Distribution by PTSD severity[1]: At MAT intake** | Asymptomatic | 0 | N/A | NA | MAPS Phase 2 trials data: Weathers 2011 |
| | Mild | 0 | | | |
| | Moderate | 122 | | | |
| | Severe | 311 | | | |
| | Extreme | 568 | | | |
| **Distribution by PTSD severity: At primary MAT follow-up** | Asymptomatic | 216 | N/A | NA | MAPS Phase 2 trials data: Weathers 2011 |
| | Mild | 270 | | | |
| | Moderate | 135 | | | |
| | Severe | 230 | | | |
| | Extreme | 149 | | | |
| **Effectiveness** | Mean change in CAPS score | -37.9 | 28.9 | Normal | MAPS Phase 2 trials data |
| **Intervention costs** | Therapists | $6,194 | N/A | N/A | CMS; Fair Health; MAPS accounting data |
| | Screening and diagnostics | $997 | | | |
| | Pharmaceutical (125 mg MDMA) | $353 | | | |
| | Total | $7,543 | +/- 30% | Beta | |
| **Health care cost (annual)** | Asymptomatic | $4,946 | $0 | Gamma | Ivanova 2011, Marciniak 2005, Chan 2009, Lavelle 2018 and authors' construction |
| | Mild | $9,944 | $0 | | |
| | Moderate | $14,916 | $0 | | |
| | Severe | $19,888 | $0 | | |
| | Extreme | $23,866 | $0 | | |
| **Mortality, relative risk** | Asymptomatic | 1.00 | N/A | Lognormal | Ahmadi 2011 and authors' construction |
| | Mild | 1.74 | 0.70 | | |
| | Moderate | 2.05 | 0.80 | | |
| | Severe | 2.51 | 1.10 | | |
| | Extreme | 2.76 | 1.25 | | |
| **Health state utilities** | Asymptomatic | 1 | N/A | Beta | Whiteford 2013, Mancino 2006 and authors' construction |
| | Mild | 0.97 | 0.9–1.0 | | |
| | Moderate | 0.851 | +/- 10% | | |
| | Severe | 0.477 | | | |
| | Extreme | 0.369 | | | |
| **Other inputs** | Cohort size | 1000 | N/A | N/A | N/A |
| | Risk of progression (annual) after year 5 | 5% | 0–10% | Beta | Authors' construction |
| | Background mortality | 0.00139 at age 40 | Age-dependent | N/A | U.S. Life-tables; National Vital Statistics Reports; https://www.cdc.gov/nchs/data/nvsr/nvsr59/nvsr59_09.pdf |
| | Discount rate | 3.0% | 2.3% - 3.8% | Beta | World Bank, 1993 |
| | Time horizon | 30 | 1–40 | N/A | Authors' construction |
| | Mean age | 40 | 35–45 | Beta | Mithoefer 2019 |

1. CAPS (revised): 0–19: asymptomatic/few; 20–39: mild PTSD/subthreshold; 40–59: moderate PTSD/threshold; 60–79 = severe, > = 80 extreme PTSD symptomatology; Weathers 2011.

benefits as discussed above. Because shorter horizons yield more reliable results, we also report on results at years 1 and 10 and find the cost breakeven point (3.1 years).

### Outcomes

Our analysis yields estimates of the cost of MAP; net costs or savings (MAP cost adjusted for future medical care costs); premature deaths averted; QALYs gained; and, in scenarios that are not cost-saving, cost per QALY gained.

### Sensitivity and scenario analysis

We conducted extensive one-, two- and multivariable sensitivity analyses to assess variation in findings given parameter value uncertainty. Sensitivity ranges for deterministic analyses were informed by the low and high estimates (typically 95% confidence intervals) reported in relevant literature. For probabilistic sensitivity analyses, we ran 10,000 Monte Carlo simulations with beta distributions specified for probabilities, gamma distributions for costs, and lognormal distributions for relative risks. MAP effectiveness in the form of changes in global CAPS-IV scores in the phase 2 trials is normally distributed. We specified distribution parameters such that the central tendencies approximate those reported in the source literature when such information was available. In a scenario analysis we projected MAP costs and benefits assuming that patients progressed to the next most severe stage of PTSD (e.g., from moderate to severe) starting in year 5 after MAP, at the rate of 6% annually (range 0% to 12%); those with extreme disease progress no further.

## Results

### Base-case results

**Intervention cost.**  The cost of the MAP intervention was $7,543 per patient who initiated the protocol. Of this, 91.2% is therapists' compensation. The remainder is the cost of MDMA, 4.7%; test kits for pregnancy and drugs of abuse, 3.7%; and nuclear stress tests, and carotid ultrasound for patients who require them, 0.4%.

**Net costs, QALYs gained and cost-effectiveness.**  Projected for 30 years, for a cohort of 1,000 patients, MAP as compared with controls averts 42.9 undiscounted deaths (90% CI, 11.1, 84.1); generates 5,553 discounted QALYs (90% CI, 4,725, 6,407); and saves a discounted net $103 million (90% CI, $71.2 - $144.3 million) in combined mental health and general medical care costs. (See S2 Fig in S1 File). As shown in Table 2, MAP is dominant (better and cheaper) unless it is assumed that benefits cease after one year following MAP, in which case the ICER is $26,427 per QALY gained. Using a 10-year horizon, MAP saves 2,517 QALYs, averts 18.9 deaths, and saves $36.7 million. MAP breaks even in costs at 3.1 years at which point it generates 918 QALYs and averts 5.9 deaths.

### Sensitivity and scenario analyses

In one-way sensitivity analyses, net costs are driven primarily by the cost of treating PTSD followed by the variation in intervention effectiveness; higher values are associated with greater net savings (Fig 2).

MAP ceases to yield net savings if the medical care costs associated with PTSD are 19.3% or lower of base-case values. At 10% of base case PTSD medical costs, MAP has an ICER of $2,124 per QALY gained (not shown). Variation in effectiveness has the greatest down-side effect on the magnitude of savings. At the low end of the 95% CI for this variable (mean reduction in CAPS-IV score of 31.3 compared with the base-case value of 37.9), net savings approximate $80 million.

**Table 2. Net present costs, health benefits and cost-effectiveness results for 30, 10, 3.1 and 1-year analytic time horizons for 1,000 patients.**

| | | | MAP | Control |
|---|---|---|---|---|
| **Intervention costs and discounted future medical care costs** | 30 years | Costs | $270,195,980 | $373,351,216 |
| | | Net cost (savings) | **($103,155,236)** | |
| | 10 years | Costs | $140,326,654 | $176,983,925 |
| | | Net cost (savings) | **($36,657,271)** | |
| | 3.1 years[a] | Costs | 61,210,801 | 61,210,801 |
| | | Net cost (savings) | **$0** | |
| | 1-year | Costs | $28,388,045 | $20,779,355 |
| | | Net cost (savings) | **$7,608,691** | |
| **Health benefits** | 30 years | QALYs | 13,591 | 8,037 |
| | | QALYs gained | **5,553** | |
| | | Deaths | 278.6 | 321.5 |
| | | Deaths averted[b] | **42.9** | |
| | 10 years | QALYs | 6,315 | 3,798 |
| | | QALYs gained | **2,517** | |
| | | Deaths | 53.4 | 72.3 |
| | | Deaths averted[2] | **18.9** | |
| | 3.1 years[a] | QALYs | 2,331.2 | 1,412.7 |
| | | QALYs gained | **918.4** | |
| | | Deaths | 14.2 | 20.1 |
| | | Deaths averted[b] | **5.9** | |
| | 1-year | QALYs | 733 | 445 |
| | | QALYs gained | **288** | |
| | | Deaths | 4.3 | 6.3 |
| | | Deaths averted[b] | **2.0** | |
| **Cost-effectiveness** | 30 years | Net cost per QALY gained | Dominant[c] | |
| | 10 years | | Dominant[c] | |
| | 3.1 years[a] | | Dominant[c] | |
| | 1-year | | **$26,427** | |

a. Approximate analytic horizon at which net costs are zero, i.e.' break-even'.

b. Undiscounted.

c. MAP is less costly and yields more QALYs; no cost-effectiveness ratio calculated.

Fig 3 shows the result of 10,000 iterations of a simulation in which effectiveness is varied across its normally-distributed range around the base-case value. Net savings are preserved across all iterations and 45.2% show savings exceeding basecase results of $103 million.

Fig 4 shows the results of a scenario analysis allowing for annual disease progression ranging from 0.0–0.12 after year 5. In 10,000 simulation iterations none shows a net cost, and 44.9% show net savings exceeding $81 million.

In a two-way sensitivity analysis, the per-patient cost of MAP is varied between $4,000 and $20,000 (base-case, $7,543) and the analytic horizon varies from 5–40 years. Using a five-year horizon and a MAP cost of $20,000, MAP breaks even at a cost of $16,232 per patient and has an ICER of $2,779 per QALY gained. (See S3 Fig in S1 File).

## Discussion

MDMA-assisted psychotherapy (MAP) provided to severely-affected patients of the type served in MAPS' phase 2 trials appears to be cost-saving, and thus highly cost-effective. For

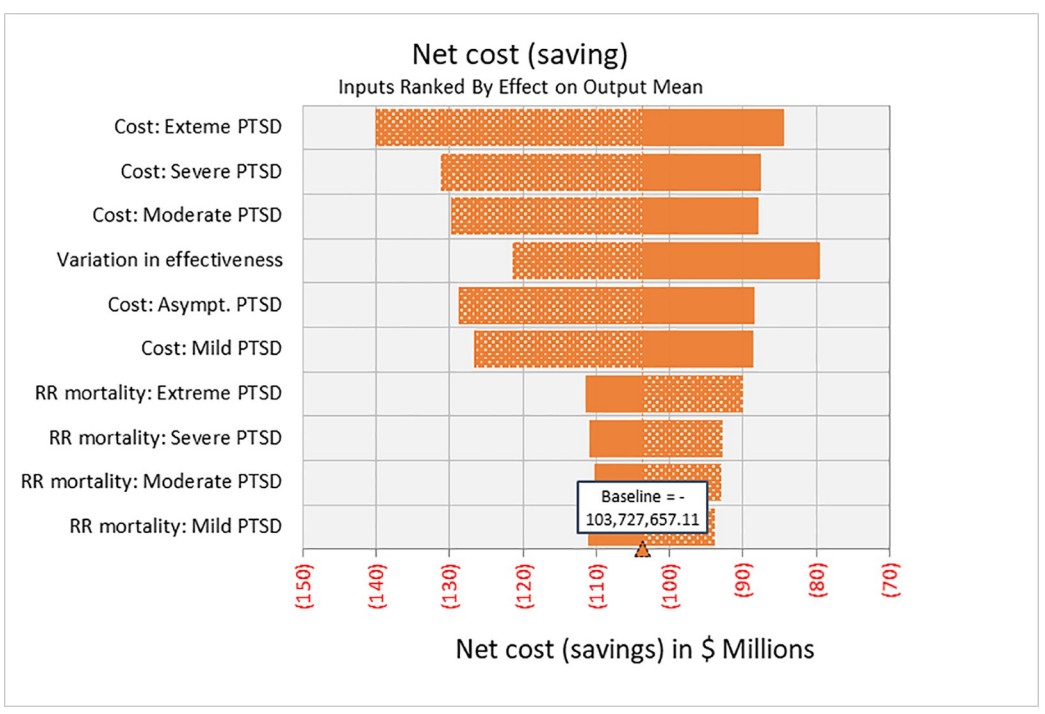

**Fig 2. One-way sensitivity analyses.** Net cost of MAP over 30-year analytic time horizon; for 1,000 patients using 10 most influential input variables across the range of values as shown in Table 1.

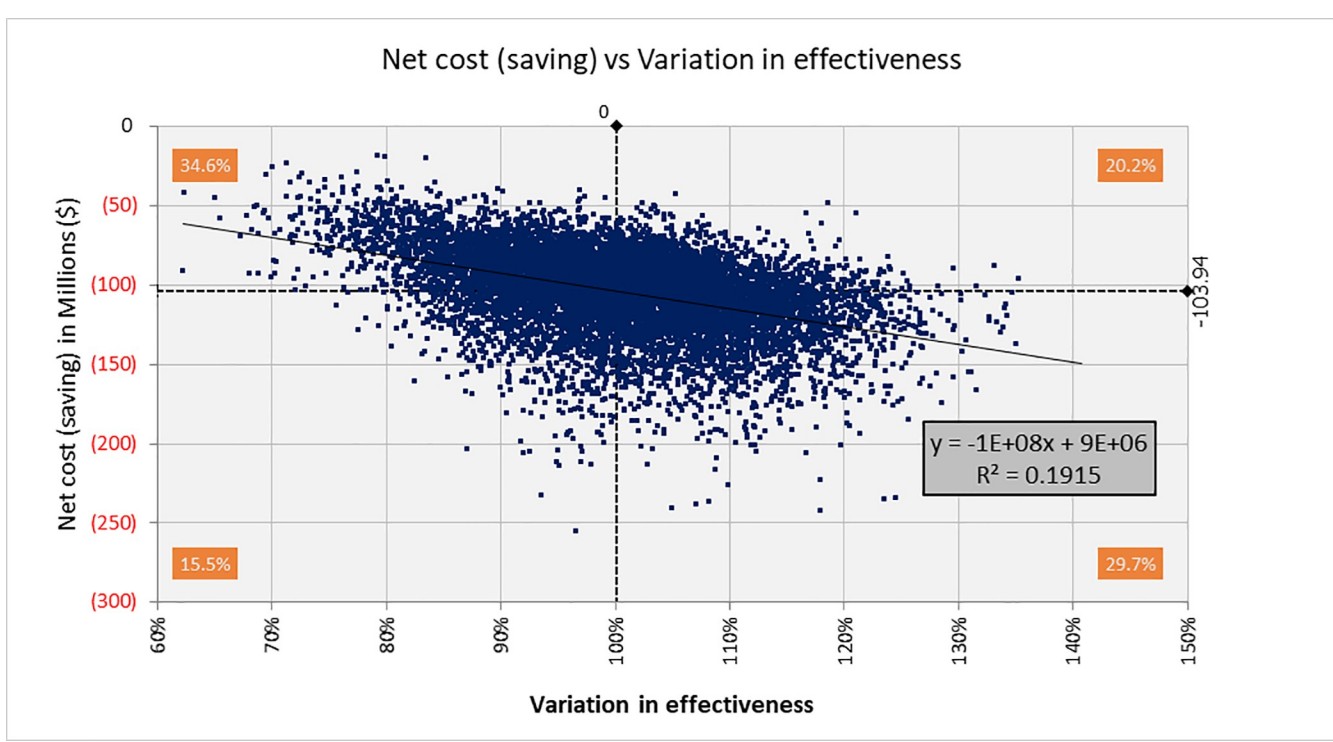

**Fig 3. One-way sensitivity analysis of MAP effectiveness on net cost (savings).** 10,000 iterations of a Monte Carlo simulation; 1,000 patients over a 30-year analytic horizon. Effectiveness varied by 1 SD of base-case value.

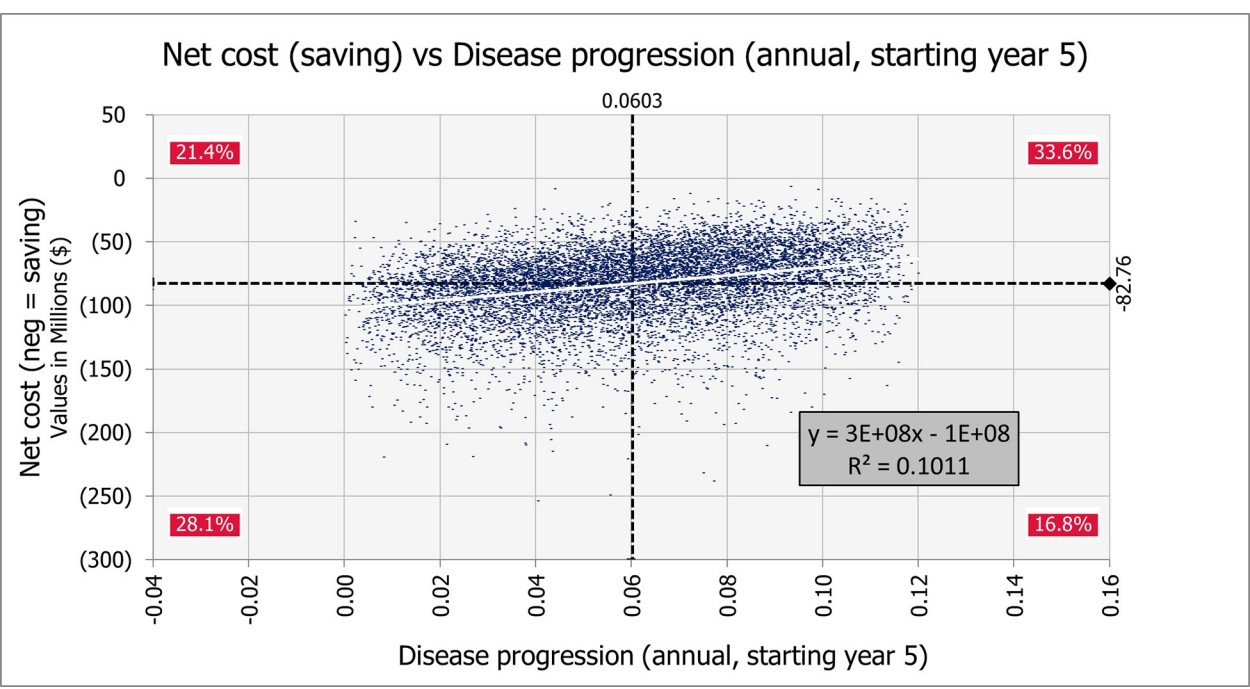

**Fig 4. One-way sensitivity analysis of disease progression on net cost (savings) for 10,000 iterations of a Monte Carlo simulation.** 10,000 iterations of a Monte Carlo simulation; 1,000 patients over a 30-year analytic horizon. Disease progression varied from 0.00 to 0.12 annually.

every 1,000 patients treated we estimate 30-year savings to the medical care system of $103 million, while generating 5,553 QALYs and averting 42.9 deaths. The finding of net savings holds across a wide range of plausible assumptions and input values.

Many third-party payers are likely to save money by including MAP as a covered benefit for patients with chronic PTSD. How any particular payer fares depends on the specific costs they face and the possibility of losing members after paying the up-front costs of MAP but before recovering costs in the form of substantial decreased utilization. There are at least two reasons for believing that these risks are modest for many payers. First, even if MAP costs considerably more than our estimates, it remains cost-saving. Second, since the break-even time is 3.1 years, patients who receive MAP would need to migrate to another plan at an average annual rate of 33% for the payer to incur net costs. In the private health insurance market patients enrolled in Exclusive, Provider Plans, Preferred Provider Plans and Health Maintenance Organizations, had plan switching rates of 27%, 24% and 13%, respectively in 2015, though rates are higher in HSA plans, 51% [33]. Medicaid enrollees have rates of coverage disruption of 23.8% in the Affordable Care Act "non-expansion" states, and 9.7% in the expansion states [34]. By contrast, such "churn" is far lower in the Veterans Affairs Health System, which cares for many ex-military patients with PTSD.

Among the strengths of this study is our ability to draw on a substantial body of RCT data on clinical effectiveness, and to portray the effects of varying assumptions regarding analytic horizon, treatment effectiveness, treatment costs and future medical costs. However, there are a number of limitations, notably both model and data uncertainty. The model excludes a number of important potential benefits to the families of PTSD patients and to the broader society. These include reduced risks and severity of substance abuse, domestic violence, and involvement with the criminal justice system [35]. In a study of Vietnam-era veterans, PTSD was associated with lower likelihood of employment and lower wages for those who are employed [36].

In an unplanned observation in the first MAPS phase 2 study, the three patients unable to work due to PTSD were able to return to work post-study [37]. Reduction of PTSD symptoms may therefore lower disability payments and raise productivity. From a societal perspective, we thus under-estimated benefits of the intervention, perhaps substantially.

Our approach projects costs and benefits beyond the four-year period for which follow-up data are available. For example, in the base-case we assume no relapse, yet it is possible that starting in year five or later, that patients do relapse. MAPS is currently planning a 5 to 15-year follow-up study of patients from the phase trials 2 to shed further light on the durability of MAP benefits and on health-care utilization.

The data on medical care costs associated with PTSD are highly variable, and the patient populations that they reflect imperfectly match those of the MAPS trials. However, the extensive sensitivity and scenario analyses provide reasonable assurance that even with less favorable values, MAP would retain attractive cost-effectiveness, and likely cost savings. Finally, our data pertain to a limited portion of PTSD patients in the U.S. It is thus not possible to generalize our results to those suffering from less severe or chronic forms of the condition. Further, the MAPS trials were designed to help those with treatment-resistant PTSD. If inclusion criteria are relaxed to include those who benefit from conventional therapies as in MAPS' phase 3 trials, incremental effectiveness may be lower; although it is also possible that those with less severe PTSD respond even better to MAP.

We are aware of no other studies of the cost-effectiveness of MAP. Thus, these are early days in understanding the costs and consequences of this novel therapy. Among the questions on the research agenda are whether it is possible to reduce costs by conducting at least some of the "pre", "post" or active MDMA psychotherapy sessions in a group context. Similarly, without sacrificing benefits, might at least one of the attending therapists be a master's level practitioner with specialized additional training such as that offered by the California Institute of Integral Studies, (https://www.ciis.edu/), rather than an MD or PhD-level practitioner? Might full safety and efficacy be retained if the second facilitator were an unlicensed student or nurse? Finally, it would be useful to compare the effectiveness and incremental cost-effectiveness of a protocol with two active sessions with that of three sessions, since there is an early suggestion of enhanced benefit with three sessions [19].

A substantial body of clinical evidence demonstrates that MDMA-assisted psychotherapy can durably alleviate PTSD symptoms in a large portion of patients with the most severe and treatment-resistant forms of the disorder. This is the first study to estimate the cost-effectiveness of this novel treatment. There is reason to be encouraged that accelerated access to this therapy could be an excellent use of health care resources.

## Supporting information

**S1 File.**
(DOCX)

## Acknowledgments

We wish to acknowledge Lia Mix for her help in costing the intervention, including identifying CPT codes, Daniel Lawhon for his capable research support and Dave Snyder for his help in keeping the lead author on schedule.

## Author Contributions

**Conceptualization:** Elliot Marseille, James G. Kahn, Berra Yazar-Klosinski, Rick Doblin.

**Data curation:** Elliot Marseille, Berra Yazar-Klosinski, Rick Doblin.

**Formal analysis:** Elliot Marseille, James G. Kahn.

**Methodology:** Elliot Marseille, James G. Kahn.

**Project administration:** Elliot Marseille.

**Resources:** Berra Yazar-Klosinski.

**Validation:** Elliot Marseille, Berra Yazar-Klosinski, Rick Doblin.

**Visualization:** Elliot Marseille, James G. Kahn.

**Writing – original draft:** Elliot Marseille.

**Writing – review & editing:** Elliot Marseille, James G. Kahn, Berra Yazar-Klosinski, Rick Doblin.

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
