## [Decision Letter · Decision Letter 0]

8 Sep 2020

PONE-D-20-21758

The cost-effectiveness of MDMA-assisted psychotherapy for the treatment of chronic, treatment-resistant PTSD

PLOS ONE

Dear Dr. Marseille,

Thank you for submitting your manuscript to PLOS ONE. After careful consideration, we feel that it has merit but does not fully meet PLOS ONE’s publication criteria as it currently stands. Therefore, we invite you to submit a revised version of the manuscript that addresses the points raised during the review process.

We look forward to receiving your revised manuscript.

Kind regards,

Stephan Doering, M.D.

Academic Editor

PLOS ONE

Reviewers' comments:

Reviewer's Responses to Questions

**Comments to the Author**

1. Is the manuscript technically sound, and do the data support the conclusions?

Reviewer #1: Yes

Reviewer #2: Yes

2. Has the statistical analysis been performed appropriately and rigorously? 

Reviewer #1: Yes

Reviewer #2: Yes

3. Have the authors made all data underlying the findings in their manuscript fully available?

Reviewer #1: Yes

Reviewer #2: Yes

4. Is the manuscript presented in an intelligible fashion and written in standard English?

Reviewer #1: Yes

Reviewer #2: Yes

5. Review Comments to the Author

Reviewer #1: This paper represents an important part of the program of work carried out by MAPS into MDMA-assisted psychotherapy. Along with understanding the safety and efficacy of this treatment approach, we need to understand the financial implications. The paper is clear and well written, and the authors have structured it well so it takes the reader logically through each step of the process. I really enjoyed reading it.

My only small comment is about Table 2 on page 15. By the time I reached this point, I wasn't completely clear who the 'controls' were and had to go back a few sections to figure it out. You have explained the controls earlier in the paper, but I wonder if this could be made clearer somehow, as who you are comparing the cases with is really important. Are they MAPS studies controls, or people who have had no treatment, or people who have had one or more other treatments. This then impacts on your discussion and conclusions - that the treatment is cost effective, but relative to the controls. So I think who the controls are needs to be made a bit clearer throughout.

Reviewer #2: This is a medical-costs evaluation on a modest sample of MDMA enhanced therapy treated patients with PTSD, finding substantial benefits. They find significant benefits after 3 years. The study is highly US-centric, in a way that would be hard to understand outside of the US, and the sample is modest, and, of those with follow-up, very small. So the outcome, though impressive, is both hard to generalise and not very robust.

Abstract

Do not use ‘chronic ptsd’ – the evidence is growing for a separate chronic PTSD diagnosis or subgroup, and it will only confuse the reader if that is not what is intended

It says nowhere which country these costs are calculated for

Intro

Line about COVID seems out of place.

Do not use ‘chronic ptsd’ – the evidence is growing for a separate diagnosis or subgroup, and it will only confuse the reader if that is not what is intended

Methods

Very small number of patients – 74 only, so number in each severity category will be v small.

What about impact on comorbidities? What about cost of side-effects? What about return to work, benefits usage? These are likely to dwarf medical costs.

What is the improvement at 3.54 years?

Figure 1 – why don’t they have the option of transitioning to a less severe form of PTSD? Though spontaneous improvement may be unlikely, continued improvement following treatment is a different question.

Results

Medical Care costs – how can moderate by 75% lower and mild 50% lower?

Table 1 – what are ‘utilities’?

Need to define acronyms – ICER, CPT, HSA

6. PLOS authors have the option to publish the peer review history of their article (what does this mean?). If published, this will include your full peer review and any attached files.

Reviewer #1: **Yes: **Louise Morgan

Reviewer #2: No

---

## [Author Response · Author response to Decision Letter 0]

12 Sep 2020

Dear Plos One Editors,

We appreciate the thoughtful comments of both reviewers and believe that the paper is now substantially clearer. Each of the reviewers’ comments is considered in turn, with corresponding changes in the manuscript noted. Please also see the revised paper with all changes showing in “Track Changes”.

Reviewer #1

Comment: My only small comment is about Table 2 on page 15. By the time I reached this point, I wasn't completely clear who the 'controls' were and had to go back a few sections to figure it out. You have explained the controls earlier in the paper, but I wonder if this could be made clearer somehow, as who you are comparing the cases with is really important. Are they MAPS studies controls, or people who have had no treatment, or people who have had one or more other treatments. This then impacts on your discussion and conclusions - that the treatment is cost effective, but relative to the controls. So I think who the controls are needs to be made a bit clearer throughout.

Response: We could not agree more. It is essential that the control condition is clearly described. The control condition in these trials was ‘active’, i.e., the intensive psychotherapy both before and after the administration of the placebo or low-dose MDMA has a significant clinical benefit. However, psychotherapy wrapped around administration of a pill known by the patient to be a placebo (or very low dose MDMA) cannot be a real-world treatment option. For the main analysis, we therefore compared the treatment group at follow-up against the control group at baseline, i.e., had the control group subjects did not receive any new intervention. This is the appropriate ‘real-world’ comparison in that he control group is assumed to continue on the same treatment trajectory that they would have followed in the absence of MDMA, or indeed, any clinical trial. In an exploratory scenario analysis, we nevertheless modelled the MDMA group against the control group at follow-up; that is, where the improvement experienced by the control group from the psychotherapy plus placebo is compared against the improvement experienced by the MDMA treatment group. As shown in the section headed “Alternative comparison: Phase 2 trial active group versus control group at primary endpoint”, in “Supplementary Materials”, the net discounted savings would be ~$79 million for 1,000 patients in each arm, with a gain of 3,985 QALYs; as opposed to savings of ~$103 million and a gain of 5,553 QALYs for the main analysis. Thus, with either approach MAP is the dominant option.

Previous version: “The control condition in the phase 2 trials does not represent a feasible, real-world treatment option since it consists of psychotherapy combined with either a placebo (two trials) or a 25-40 mg dose of MDMA (four trials) believed to be clinically inactive (18). In the base-case analysis, we therefore modeled the costs and benefits of the active treatment group after receiving MAP with the same group at baseline, i.e., as if they had not received MAP. Because those in the control condition experienced some improvement, in a scenario analysis an additional comparison was implemented, namely of patients in the active treatment arm versus controls. (See Supplement, “Alternative comparison” and “Further evidence of MAP effectiveness”).”

Current version: “The control condition in the phase 2 trials does not represent a feasible treatment option since it consists of psychotherapy combined with either a placebo (two trials) or a 25-40 mg dose of MDMA (four trials) believed to be clinically inactive (18). In order to portray a realistic comparison to MAP, one that reflects the ‘real-world’ treatment alternative, we modeled the costs and benefits of the active treatment group after receiving MAP with the same group at baseline, i.e., as if they had not received MAP. Because those in the control condition experienced some improvement, perhaps from the intensive course of psychotherapy that patients in both arms received, in a scenario analysis, an alternative comparison was implemented, namely of patients in the treatment versus control arm at follow-up. (See Supplement, “Alternative comparison”).”

Reviewer #2

ABSTRACT

Comment: Do not use ‘chronic ptsd’ – the evidence is growing for a separate chronic PTSD diagnosis or subgroup, and it will only confuse the reader if that is not what is intended.

Response: We are unsure of the context of this question, and in particular of emerging evidence for a ‘chronic’ subgroup that differs in definition from our use of the word. By study inclusion criteria, subjects were enrolled only if they had PTSD for six months or longer. The DSM-4 definition of chronic PTSD is 'greater than or equal to 3 months' symptom duration. In our studies we extended this to 6 months in order to be sure that the chance of natural recovery from PTSD was further limited. In fact, the actual duration of symptoms of the patients in the trial was around 200 months on average. However, we agree that we should be explicit about this definition and have therefore amended both the Abstract and Methods sections as follows. 

In Abstract

Previous version: “Efficacy was based on the pooled results of six randomized controlled phase 2 trials with 105 subjects; and a four-year follow-up of 19 subjects.”

Amended version: “Efficacy was based on the pooled results of six randomized controlled phase 2 trials with 105 subjects (average duration of PTSD, 197.9 months (SD, 139.1) and 222.6 months (SD, 208.5) in controls and treatment groups respectively); and a four-year follow-up of 19 subjects.”

In Methods

Previous version: Patient population. The 105 subjects of the six double-blind controlled pilot studies (31 control; 74 active group) suffered moderate to extreme chronic PTSD with an average duration of 197.9 months (SD, 139.1) in the controls and 222.6 months (SD, 208.5) in the treatment group. 

Amended version: The 105 subjects of the six double-blind controlled pilot studies (31 control; 74 active group) suffered moderate to extreme chronic PTSD. “Chronic” was defined in these trials as a minimum duration of symptoms of six months. The average duration was 197.9 months (SD, 139.1) in the controls and 222.6 months (SD, 208.5) in the treatment group.

We hope that this provides sufficient clarification. 

Comment: It says nowhere which country these costs are calculated for

Response: Good point.

In Abstract

Previous version: To assess the cost-effectiveness of MDMA-assisted psychotherapy (MAP) from the health care payer’s perspective, we constructed a decision-analytic Markov model to portray the costs and health benefits of treating patients with chronic, severe, or extreme, treatment-resistant PTSD with MAP.

Amended version: To assess the cost-effectiveness of MDMA-assisted psychotherapy (MAP) from the U.S. health care payer’s perspective, we constructed a decision-analytic Markov model to portray the costs and health benefits of treating patients with chronic, severe, or extreme, treatment-resistant PTSD with MAP.

In Methods

Previous version: We estimated the all-cause medical costs of patients with PTSD from four studies with five separate estimates (28-31).

Amended version: Medical care costs. We estimated the all-cause medical costs borne by U.S. medical care payers of patients with PTSD from four studies with five separate estimates (28-31).

INTRO

Comment: Line about COVID seems out of place.

Response: Agree. We have now deleted that sentence.

Previous version: “It can also cause stress-mediated physical health problems such as cardiovascular disease and type-2 diabetes (5), alcohol abuse, high caloric intake and BMI, and smoking (6-8). Early indications suggest that COVID-19 may cause PTSD in young people (9). In addition to burden of disease, PTSD generates substantial medical care costs in the U.S. including $44.3 billion (2019 dollars) for hospitalization between 2002-2011 (1).” 

Amended version: “It can also cause stress-mediated physical health problems such as cardiovascular disease and type-2 diabetes (5), alcohol abuse, high caloric intake and BMI, and smoking (6-8). In addition to burden of disease, PTSD generates substantial medical care costs in the U.S. including $44.3 billion (2019 dollars) for hospitalization between 2002-2011 (1).” 

METHODS

Comment: Very small number of patients – 74 only, so number in each severity category will be v small.

Response: Our analysis is based on 105 subjects in total: 31 controls and 74 in the treatment group. All else equal, larger study populations are always welcome. Nevertheless, even with this, moderate-sized population, the p-value for the main efficacy outcome was <0.0001. 

Comment: What about impact on comorbidities? What about cost of side-effects? What about return to work, benefits usage? These are likely to dwarf medical costs.

Response: Thank you for raising these issues Each is considered in turn:

Impact on comorbidities: Because PTSD is associated with so many other adverse health conditions this is an important issue, and one which we believe is dealt with effectively in the article. The medical costs included in the model consider all causes, not just those directly associated with PTSD. However, we particularly thank the reviewer for this comment because we did not make this sufficiently clear. We have added the phrase “all-cause” in both the Abstract and in the “Medical care costs” paragraph. 

Side-effect: Assessing the costs and ‘disutility’ of drug side-effects is often an important part of pharmacoeconomic analyses. We state in the “Health state and utility values” paragraph, “Adverse events associated with the trial were transient and managed with benzodiazepines or sleep aids, and are not reflected in the current analysis (17).” The duration of these effects are typically hours, and the cost of benzodiazepines, for example, are a negligible fraction of the total intervention cost. In the context of the decades-long time horizon for both health outcomes and medical care costs, quantifying these effects would have no appreciable influence on our findings. They would in fact introduce false precision in our results. 

Return to work. As the reviewer points out, the effect of MAP on productivity could be important. This is a complex analysis in its own right and would be part of a broader assessment of the societal impact of MAP. As stated in the Abstract we conducted this analysis from the health care payer’s perspective. As we point out in the Discussion section, including productivity effects would serve to make MAP even more cost-saving: “Reduction of PTSD symptoms may therefore lower disability payments and raise productivity. From a societal perspective, we thus under-estimated benefits of the intervention, perhaps substantially.” It would not qualitatively change our main findings. 

Comment: What is the improvement at 3.54 years?

Response: Quoting from the relevant study (Mithoefer 2013): “Results for the 16 CAPS completers showed there were no statistical differences between mean CAPS score at LTFU (mean = 23.7; SD = 22.8) (tmatched = 0.1; df = 15, p = 0.91) and the mean CAPS score previously obtained at Study Exit (mean = 24.6, SD = 18.6).” As is stated in the foregoing, and as we stated in our paper in the “Representation of clinical trial result” paragraph, this is “ . . . a modest though statistically insignificant improvement in PTSD . . .” Because it is not statistically significant, we would prefer not to give the details mentioned above regarding the change in CAPS scores. We fear it will distract from the real point we want to make.

Comment: Figure 1 – why don’t they have the option of transitioning to a less severe form of PTSD? Though spontaneous improvement may be unlikely, continued improvement following treatment is a different question.

Response: This is an important point. For the MDMA treatment group, we know that there is no statistically significant improvement for at least ~4 years post-treatment, and this is currently described in the paper. (See response to previous comment). There are no empirical data bearing on this point beyond that time horizon. However, it is consistent with a ‘conservative’ (meaning unfavorable for MDMA cost-effectiveness) analysis to assume no further improvement in this group. For the control group, the evidence is fairly strong for assuming no further improvement in this class of treatment-resistant, severely-affected patients who have had PTSD for over 200 months on average. This is described, with citations on the bottom half of the paragraph on “Representation of clinical trial results”. 

RESULTS

Comment: Medical Care costs – how can moderate by 75% lower and mild 50% lower?

Response: Good point. This was bad wording on our part. We now say, “The cost associated with extreme PTSD is 20% higher than severe PTSD whereas moderate and mild are 75% and 50% of the cost of severe, respectively (subjected to sensitivity analyses)”. 

Comment: Table 1 – what are ‘utilities’?

Response: In the text of the paper we refer to “Health state utilities” and have amended Table 1 to use the same terminology.

Comment: Need to define acronyms – ICER, CPT, HSA

Response: Thank you for catching this. We have replaced the first instance of each acronym with the full text needed.

---

## [Editor Report · Decision Letter 1]

17 Sep 2020

The cost-effectiveness of MDMA-assisted psychotherapy for the treatment of chronic, treatment-resistant PTSD

PONE-D-20-21758R1

Dear Dr. Marseille,

We’re pleased to inform you that your manuscript has been judged scientifically suitable for publication and will be formally accepted for publication once it meets all outstanding technical requirements.

Kind regards,

Stephan Doering, M.D.

Academic Editor

PLOS ONE

---

## [Editor Report · Acceptance letter]

29 Sep 2020

PONE-D-20-21758R1 

The cost-effectiveness of MDMA-assisted psychotherapy for the treatment of chronic, treatment-resistant PTSD 

Dear Dr. Marseille:

I'm pleased to inform you that your manuscript has been deemed suitable for publication in PLOS ONE. Congratulations! Your manuscript is now with our production department. 

Kind regards, 

on behalf of

Professor Stephan Doering 

Academic Editor

PLOS ONE